# Dowker complex based machine learning (DCML) models for protein-ligand binding affinity prediction

Xiang Liu[1,2,3], Huitao Feng[2,4], Jie Wu[3,5], Kelin Xia[1]*

**1** Division of Mathematical Sciences, School of Physical and Mathematical Sciences, Nanyang Technological University, Singapore, **2** Chern Institute of Mathematics and LPMC, Nankai University, Tianjin, China, **3** Center for Topology and Geometry Based Technology, Hebei Normal University, Hebei, China, **4** Mathematical Science Research Center, Chongqing University of Technology, Chongqing, China, **5** School of Mathematical Sciences, Hebei Normal University, Hebei, China

* xiakelin@ntu.edu.sg

**Data Availability Statement:** The PDBbind datasets are available at http://www.pdbbind.org.cn/, and the code is available on GitHub at https://github.com/LiuXiangMath/Dowker-Complex-Based-ML.

## Abstract

With the great advancements in experimental data, computational power and learning algorithms, artificial intelligence (AI) based drug design has begun to gain momentum recently. AI-based drug design has great promise to revolutionize pharmaceutical industries by significantly reducing the time and cost in drug discovery processes. However, a major issue remains for all AI-based learning model that is efficient molecular representations. Here we propose Dowker complex (DC) based molecular interaction representations and Riemann Zeta function based molecular featurization, for the first time. Molecular interactions between proteins and ligands (or others) are modeled as Dowker complexes. A multiscale representation is generated by using a filtration process, during which a series of DCs are generated at different scales. Combinatorial (Hodge) Laplacian matrices are constructed from these DCs, and the Riemann zeta functions from their spectral information can be used as molecular descriptors. To validate our models, we consider protein-ligand binding affinity prediction. Our DC-based machine learning (DCML) models, in particular, DC-based gradient boosting tree (DC-GBT), are tested on three most-commonly used datasets, i.e., including PDBbind-2007, PDBbind-2013 and PDBbind-2016, and extensively compared with other existing state-of-the-art models. It has been found that our DC-based descriptors can achieve the state-of-the-art results and have better performance than all machine learning models with traditional molecular descriptors. Our Dowker complex based machine learning models can be used in other tasks in AI-based drug design and molecular data analysis.

## Author summary

With the ever-increasing accumulation of chemical and biomolecular data, data-driven artificial intelligence (AI) models will usher in an era of faster, cheaper and more-efficient drug design and drug discovery. However, unlike image, text, video, audio data, molecular

**Funding:** This work was supported in part by Nanyang Technological University Startup Grant M4081842 and Singapore Ministry of Education Academic Research fund Tier 1 RG109/19, MOE-T2EP20120-0013 and MOE-T2EP20220-0010. The first author (XL) was supported by Nankai Zhide foundation. The second author (HF) was supported by Natural Science Foundation of China (NSFC grant no. 11931007, 11221091, 11271062, 11571184). The third author (JW) was supported by Natural Science Foundation of China (NSFC grant no. 11971144) and High-level Scientific Research Foundation of Hebei Province. The funders had no role in study design, data collection and analysis, decision to publish, or preparation of the manuscript.

**Competing interests:** The authors have declared that no competing interests exist.

data from chemistry and biology, have much complicated three-dimensional structures, as well as physical and chemical properties. Efficient molecular representations and descriptors are key to the success of machine learning models in drug design. Here, we propose Dowker complex based molecular representation and Riemann Zeta function based molecular featurization, for the first time. To characterize the complicated molecular structures and interactions at the atomic level, Dowker complexes are constructed. Based on them, intrinsic mathematical invariants are derived and used as molecular descriptors, which can be further combined with machine learning and deep learning models. Our model has achieved state-of-the-art results in protein-ligand binding affinity prediction, demonstrating its great potential for other drug design and discovery problems.

## Introduction

Featurization (or feature engineering) is of essential importance for AI-based drug design. The performance of quantitative structure activity/property relationship (QSAR/QSPR) models and machine learning models for biomolecular data analysis is largely determined by the design of proper molecular descriptors/fingerprints. Currently, more than 5000 types of molecular descriptors, which are based on molecular structural, chemical, physical and biological properties, have been proposed [1, 2]. Among these molecular features, structural descriptors are the most-widely used ones and can be classified into one-dimensional (1D), two-dimensional (2D), three-dimensional (3D), and four-dimensional (4D) [1, 2]. In general, 1D molecular descriptors are atom counts, bond counts, molecular weight, fragment counts, functional group counts, and other summarized general properties. The 2D molecular descriptors are topological indices, graph properties, combinatorial properties, molecular profiles, autocorrelation coefficients, and other topological/graphic/combintorial properties. The 3D molecular descriptors are molecular surface properties, volume properties, autocorrelation descriptors, substituent constants, quantum chemical descriptors, and other geometric or density-function related properties. The 4D chemical descriptors are usually generated from a dynamic process that covers various molecular configurations. Further, various molecular fingerprints are proposed, including substructure key based fingerprints [3], path-based fingerprints [4, 5], circular fingerprints [6], pharmacophore fingerprints [7, 8], and autoencoded fingerprints. Different from molecular descriptors, molecular fingerprint is large-sized vector of molecular features that are systematically generated based on molecular properties, in particular, structural properties. Deep learning models, such as antoencoder, CNN, and GNN, have also been used in molecular fingerprint generation [9–13].

The generalizability and transferability of QSAR/QSPR and machine learning models are highly related to molecular descriptors or fingerprints. Features that characterize more intrinsic and fundamental properties can be better shared between data and "understood" by machine learning models. Mathematical invariants, from geometry, topology, algebra, combinatorics and number theory, are highly abstract quantities that describe the most intrinsic and fundamental rules and properties in nature sciences. In particular, topological and geometric invariants based molecular descriptors have achieved great successes in various steps of drug design, including protein-ligand binding affinity prediction [14–18], protein stability change upon mutation prediction [19, 20], toxicity prediction [21], solvation free energy prediction [22, 23], partition coefficient and aqueous solubility [24], binding pocket detection [25], and drug discovery [26]. These models have also demonstrated great advantages over traditional

molecular representations in D3R Grand challenge [27–29]. Recently, persistent models, including hypergraph-based persistent homology [30, 31], persistent spectral [32], and persistent Ricci curvature [33–36], have been developed for molecular characterization and delivered great performance in protein-ligand binding affinity prediction.

Dowker complex (DC) is developed for the characterization of relations between two sets [37–39]. Mathematically, Dowker complex (DC) is defined on two sets $X$ and $Y$ with a relation $R$, which is a subset of the product set $X \times Y$. The elements in the same set can form a simplex in DC if they all have relation with a common element from the other set. Note that only elements in the same set, i.e., either $X$ or $Y$, can form simplexes. Stated differently, a simplex in DC can never be formed among elements from both sets. In this way, a DC can be separated into two disjoint simplicial complexes, i.e., one with elements all from $X$ and the other with elements all from $Y$. These two simplicial complexes share the same homology groups, homotopy groups, and homotopy types [37, 38]. Moreover, DCs are equivalent to Neighborhood complex (NC) for all bipartite graphs. In fact, if the relations between two sets are represented by a bipartite graph, its associated DC is exact the same as NC. Further, Riemann Zeta function or Euler Riemann Zeta function, is a mathematical function of a complex variable. The Riemann Zeta function plays a pivotal role in analytic number theory and has applications in physics, probability theory, and applied statistics. Mathematically, Riemann Zeta function can be used in the characterization of intrinsic information of the system.

Here we propose Dowker complex based molecular interaction representations and Riemann Zeta function based molecular featurization, for the first time. More specifically, a bipartite graph can be used to model the interactions between two molecules, such as a protein and a ligand. Mathematically, a bipartite graph can be viewed as a relation between two sets, and a Dowker complex can be generated naturally from it. Further, a DC has two disjoint components, which share the same homology groups. For a protein-ligand complex, protein-based DC and ligand-based DC have the same Betti number. Further, DC-based persistent spectral models can be constructed from a filtration process, and persistent Riemann Zeta functions are used as molecular descriptors or fingerprints. Our DC-based machine learning models, in particular, DC-based gradient boosting tree (DC-GBT), are extensively tested on the three most-commonly used datasets from the well-established protein-ligand binding databank of PDBbind. It is found that our DC-GBT model has achieved state-of-the-art results and are better than all machine learning models with traditional molecular descriptors.

## Results

### DC-based biomolecular interaction analysis

Molecular representation and featurization are of essential importance for the analysis of molecular data from materials, chemistry and biology. Mathematical invariant based molecular descriptors are of greater transferability thus have achieved better performance in AI-based drug design [20, 40–42]. Here we propose the first DC-based representations for molecular interaction analysis.

**Bipartite graph-based molecular interaction characterization.**   Graph theory is widely used for the description and characterization of molecular structures and interactions. A molecular graph $G = (V, E)$ is composed of a set of vertices $V$, with each vertex representing molecular atom, residue, motif, or even the entire molecule, and a set of edges $E$, representing interactions of various kinds including covalent bonds, van der Walls, electrostatic, and other non-covalent forces. Both intra- and inter- molecular interactions, i.e., interactions within and between molecules, can be represented as bipartite graphs (also known as bigraphs or 2-mode networks). Mathematically, a bipartite graph $G(V_1, V_2, E)$ has two vertex sets $V_1$ and $V_2$, and

all its edges are formed only between the two vertex sets. Recently, bipartite-graph based inter-active matrixes have been used for machine learning models in drug design and achieved great success [20, 30–34, 40]. Mathematically, these interactive matrixes, which are based on atomic distances and electrostatic interactions, can be transformed into a weighted biadjacency matrixes between protein and ligand atoms. More specifically, if we let $V_P = \{v_i | i = 0, 1, \ldots, N_P\}$ and $V_L = \{v_j | j = 0, 1, \ldots, N_L\}$ represent the coordinate sets of protein and ligand atoms respectively, the biadjacency matrix $\mathbf{B}$ with a size $N_P \times N_L$ is defined as follows,

$$\mathbf{B}(v_i, v_j) = w_{ij}, v_i \in V_P \text{ and } v_j \in V_L. \tag{1}$$

The weights $w_{ij}$ can be chosen as the Euclidean distances or electrostatic interactions [20]. Essentially, inter-molecular interactions between protein and ligand atoms are characterized in the above biadjacency matrix.

Two individual unipartite graphs $G_1$ and $G_2$ can be constructed from a bipartite graph $G$ $(V_1, V_2, E)$ through a projection process [43, 44]. More specifically, the unipartite graph $G_1$ is generated from vertex set $V_1$, and its edges are defined between any two vertices that have a common neighborhood vertex in $V_2$. Similarly, the unipartite graph $G_2$ is based on vertex set $V_2$, and any two vertices that have a common neighborhood vertex in $V_1$ will form an edge in $G_2$. Mathematically, the connection matrixes for the unipartite graphs can be generated from the adjacency matrix as in Eq (1). The connection matrixes for protein and ligand are $\mathbf{B}\mathbf{B}^T$ and $\mathbf{B}^T\mathbf{B}$, respectively. Note that the two matrixes are of different sizes. Further, based on the uni-partite graphs $G_1$ and $G_2$, two flag complexes (or clique complexes), $K_{F,1}(G)$ and $K_{F,2}(G)$ can be constructed respectively. More specifically, in the two flag complexes, a $k$-complex is formed among $k + 1$ vertices when any two vertices are connected by an edge.

**Bipartite graph-based DC models.** Mathematically, a bipartite graph can be seen as a relation. Two Dowker complexes $K_{D,1}(G)$ and $K_{D,2}(G)$ can be naturally constructed from a bipartite graph $G(V_1, V_2, E)$. The DC $K_{D,1}(G)$ is defined on $V_1$, and its $k$-simplex is formed among $k + 1$ vertices which have "relations", i.e., forming edges, with a common vertex in $V_2$. Similarly, the DC $K_{D,2}(G)$ is based on $V_2$, and its $k$-simplex is formed among $k + 1$ vertices which are "related to" a common vertex in $V_1$. Note that when vertices are related to a com-mon vertex, that means they share the same common neighborhood vertex and a DC-based simplex will be formed among them.

Further, the Dowker theorem states that the homology of $K_{D,1}(G)$ and $K_{D,2}(G)$ are isomor-phic, which means $H_p(K_{D,1}(G)) \cong H_p(K_{D,1}(G))(p > 0)$, and 0-th homology is isomorphic if the bipartite graph $G$ is connected [38, 39]. It is worth mentioning that the flag complexes, i.e., $K_{F,1}(G)$ and $K_{F,2}(G)$, from the unipartite graphs are usually different from the DCs. Fig 1 illus-trates the bipartite graph-based DCs and their persistent barcodes. The bipartite graph is gen-erated from the phosphorus-phosphorus (P-P) interactions between two chains from DNA 1D77. The corresponding DC has two disjoint components, one from chain A and the other from chain B. The distance-based filtration process is considered and two persistent barcodes for chain A and chain B are generated. It can be observed that $\beta_1$ persistent barcodes are exactly the same. Note that $\beta_0$ persistent barcodes are not the same as the bipartite graphs are not always connected during the filtration process.

## DC-based persistent spectral models

For all the persistent models, including persistent homology/cohomology, persistent spectral and persistent function, the key point is the filtration process. There are various ways to define the filtration parameter, leading to different filtration processes. For topology-based protein-ligand interaction models, we can define the filtration parameter $f$ as the weight value of the

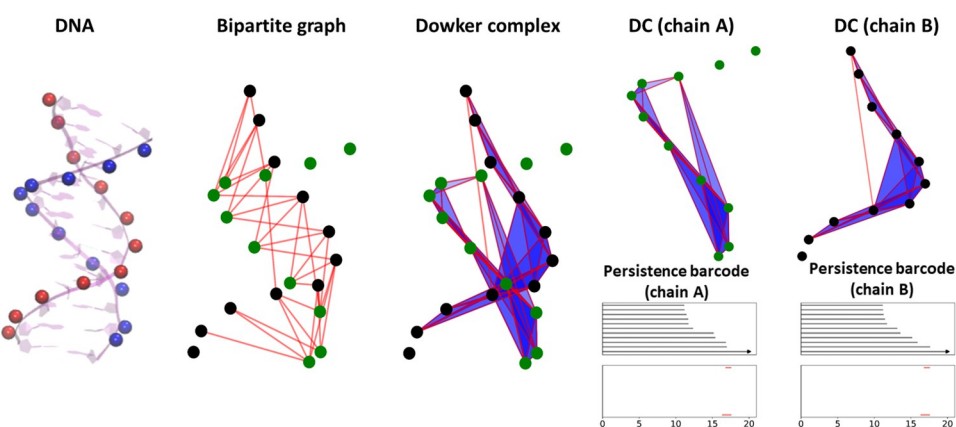

**Fig 1. Dowker complex based representation for the atomic interactions between two chains of DNA 1D77.** Only the Phosphor (P) atoms of the DNA are considered. A bipartite graph is constructed between the two DNA chains, i.e., chain A and chain B, using a cutoff distance of 16.5 Å. The corresponding Dowker complex is generated and consists of two disjoint components, one from chain A and the other from chain B. The cutoff distance can be used a filtration parameter and two persistent barcodes are obtained. It can be seen that the $\beta_1$ persistent barcodes are exactly the same for the two types of DCs from chain A and chain B. The $\beta_0$ persistent barcodes are different because the bipartite graph are not always connected during the filtration process.

biadjacency matrix in Eq (1). With the increase of filtration value, a series of nested bipartite graphs can be generated,

$$G_{f_0} \subset G_{f_1} \subset \ldots \subset G_{f_n}. \tag{2}$$

Here $(f_0 \leqslant f_1 \leqslant \ldots \leqslant f_n)$ are filtration values. The corresponding DCs can be constructed accordingly as follows,

$$K_D(G_{f_0}) \subset K_D(G_{f_1}) \subset \ldots \subset K_D(G_{f_n}). \tag{3}$$

In fact, we have two disjointed series of nested DCs as follows,

$$K_{D,1}(G_{f_0}) \subset K_{D,1}(G_{f_1}) \subset \ldots \subset K_{D,1}(G_{f_n}). \tag{4}$$

$$K_{D,2}(G_{f_0}) \subset K_{D,2}(G_{f_1}) \subset \ldots \subset K_{D,2}(G_{f_n}). \tag{5}$$

Note that the first DC series $\{K_{N,1}(G_{f_i})\}$ are for protein part, all of their vertices are protein atoms. In contrast, the second DC series $\{K_{N,2}(G_{f_i})\}$ are fully based on ligand atoms. From Dowker's theorem, these two DC series share the same homology groups, i.e., $H_p(K_{D,1}(G_{f_i})) \cong H_p(K_{D,2}(G_{f_i}))$ $(p > 0, i = 1, 2, \ldots, n)$.

Persistent spectral (PerSpect) models are proposed to study the persistence and variation of spectral information of the topological representations during a filtration process [32]. These spectral information can be used as molecular descriptors or fingerprints and combined with machine learning models for drug design. Here we study DC-bases persistent spectral models. From the Eqs (4) and (5), two sequences of Hodge Laplacian matrixes can be generated respectively (see Materials and methods). These matrixes characterize the interactions between protein and ligand atoms at various different scales. The spectral information derived from these Hodge Laplacian matrixes are used for the characterization of protein-ligand interactions. Fig 2 illustrates DC-based filtration process and the corresponding Hodge Laplacian matrixes for the protein-ligand complex 2POG. From the bipartite sequence, two separated series of DCs

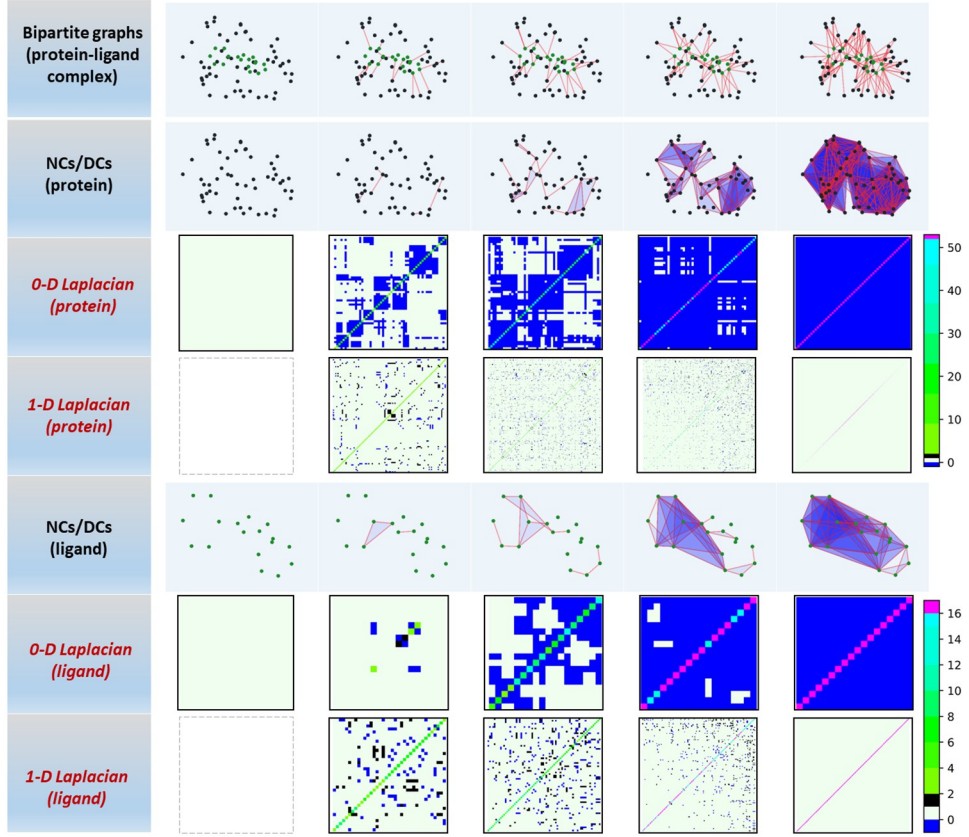

**Fig 2. Persistent combinatorial Laplacian matrixes for Dowker complex from C-C pair of PDBID 2POG.** As in the picture, based on the filtration process of bipartite graphes, a filtration of Dowker complexes can be generated and further divided into two disjoint filtration processes in protein and ligand. Then for each filtration process, two sequence of laplacian matrixes in dimension 0 and 1 are depicted. The cutoff extracting the binding core region is 5Å, filtration values are 3.5Å, 4Å, 4.2Å, 4.5Å and 5Å. For 0-D laplacian matrixes, with the increase of filtration value, the matrix size is always same, off-diagonal entries decrease from 0 to -1 until all become -1 and diagonal entries increase until all up to the number of 0-simplexes minus 1. For 1-D laplacian matrixes, the matrix size increase consistently until up to a constant, and off-diagonal entries have nonzero values 1 and -1 due to their orientation and the number of off-diagonal nonzero entries increase at early stage and then decrease until all go to zero, and diagonal entries increase until all up to the number of 0-simplexes.

are generated based on protein atoms and ligand atoms respectively. We consider the 0-dimensional (0-D) and 1-dimensional (1-D) Hodge Laplacian matrices. Note that 0-D Hodge Laplacian matrices represent topological connections between vertices, while 1-D matrixes characterize topological connections between edges. Similarly, other-higher dimensional Hodge Laplacian matrixes can be generated for higher-dimensional simplexes. Further, the multiplicity of zero-eigenvalues for the $k$-th dimensional Hodge Laplacian matrices is $\beta_k$, i.e., the $k$-th Betti number. Additionally, information from non-zero-eigenvalues indicates "geometric" properties of the simplicial complexes [32].

Spectral information, i.e., eigenvalues and eigenvectors, from our PerSpect models can not be directed used in machine learning models. This is due to the reason that their sizes vary dramatically during the filtration process. As seen in Fig 2, the number of 1-simplexes (edges) increases greatly during the filtration. In this way, the size of 1-D Hodge Laplacian matrices and the number of related eigenvalues and eigenvectors will increase with the filtration. In our PerSpect models, a series of persistent attributes are considered [32]. The persistent attributes

are statistic and combinatorial properties of eigenvalues from the sequences of Hodge Laplacian matrices. They characterize the persistence and variation of spectral information during the filtration.

Here we can use eigenvalue-based Riemann Zeta functions. More specifically, for a set of eigenvalues $\{\lambda_1, \lambda_2, \ldots, \lambda_n\}$, the Riemann Zeta functions are defined as,

$$\zeta(s) = \sum_{i=1}^{n} \frac{1}{\lambda_i^s}.$$

They can be used as molecular features for machine learning. In our model, we consider 11 types of different Riemann Zeta functions, i.e., $\zeta(s)$ with $s$ = 5, 4, 3, 2, 1, 0, -1, -2, -3, -4, -5. Note that Riemann Zeta functions are related to different persistent spectral moments.

## DC-based machine learning models for protein-ligand binding affinity prediction

The prediction of protein-ligand binding affinity is arguably the most important step in virtual screening and AI-based drug design. Here we consider DC-based machine learning models. To characterize the detailed interactions between protein and ligand atoms, we consider element-specific bipartite graph representations. More specifically, we decompose the protein atoms at binding core regions into four groups according to their atoms types, including C, N, O, and S. Ligand atoms at binding core regions are decomposed into nine groups according to their atoms types, including C, N, O, S, P, F, Cl, Br, and I. In this way, there are totally $36 = 4 \times 9$ groups of atom combinations, and protein-ligand interactions can be represented by 36 types of bipartite graphs from these atom combinations.

The bipartite graphs can be generated based on atom distances and electrostatic interactions. As stated above, the bipartite biadjacency matrix is represented as Eq (1). There are two different ways to define the weights. One is based on the Euclidian distance between atoms, that is $w_{ij} = d(v_i, v_j)$ with $d(v_i, v_j)$ the distance between atoms $v_i$ and $v_j$. The other is based on atomic electrostatic interactions, that is $w_{ij} = \frac{1}{1+exp\left(-\frac{cq_iq_j}{d(v_i,v_j)}\right)}$ with $q_i$ and $q_j$ the partial charges of atoms $v_i$ and $v_j$ and parameter $c$ a constant value (usually taken as 100). With the importance of hydrogen atom for electrostatic interactions, H atoms are usually taken into consideration and a total number of 50 types of atom combinations are considered for electrostatic interactions. The software "PDB2PQR" [45] is used to generate partial charge for protein while the partial charge of ligand can be found in PDBBind database.

We consider three most commonly-used datasets from PDBbind databank, including PDB-v2007, PDB-v2013 and PDB-v2016, as benchmark for our DC-based machine learning models. The detailed training and testing information are listed in Table 1. The binding core region is considered by using a cutoff distance of 10Å, that is all protein atoms within 10Å of any ligand atom. For distance-based DC models, the filtration goes from 2Å to 10Å with step 0.1Å, and for electrostatic-based DC models, the filtration goes from 0 to 1 with step 0.02. We only

**Table 1. Detailed information of the three PDBbind databases, i.e., PDBbind-v2007, PDBbind-v2013 and PDBbind-v2016.**

| Dataset | Refined set | Training set | Test set (Core set) |
|---|---|---|---|
| PDBbind-v2007 | 1300 | 1105 | 195 |
| PDBbind-v2013 | 2959 | 2764 | 195 |
| PDBbind-v2016 | 4057 | 3772 | 285 |

**Table 2. The parameters for our DC-based gradient boosting tree (GBT) models.**

| No. of Estimators | Learning rate | Max depth | Subsample |
|---|---|---|---|
| 40000 | 0.001 | 6 | 0.7 |
| Min_samples_split | Loss function | Max features | Repetitions |
| 2 | Least square | SQRT | 10 |

consider 0-D persistent spectral information. In this way, the size of feature vectors are 63360 = 36(atom combinations) × 80(filtration values) × 11(Riemann Zeta functions) × 2(two DCs), 55000 = 50(atom combinations) × 50(filtration values) × 11(Riemann Zeta functions) × 2(two DCs) and 118360 = 63360 + 55000 for distance-based model, electrostatic-based model and combined model respectively. Gradient boosting tree is considered to alleviate the overfitting problem. The GBT setting is listed in Table 2.

**Scoring power.** The results for our DC-GBT models are listed in Table 3. Note that 10 independent regressions are performed and the median values of Pearson correlation coefficient (PCC) and root mean square error(RMSE) are taken as the final performance of our model. Further, we systematically compare our model with existing models with traditional learning descriptors [46–54]. Detailed comparison results can be found in Fig 3. It can be seen that our model outperforms all the other machine learning models with traditional molecular descriptors, for all three datasets.

Further, we compare our DC-GBT model with advanced-mathematical based machine learning models [14, 18, 20, 30]. The results are presented in Table 4. Our DC-GBT model is ranked as second for PDBbind-v2016 dataset and PDBbind-v2013 datasets (after TopBP). Note that the accuracy of our DC-based models can be further improved if convolutional neural network models, such as the one used in TopBP models, are designed and employed.

More recently, some deep learning models for protein-ligand binding affinity prediction are proposed, such as the graphDelta model [55], ECIF model [56], OnionNet-2 model [57], DeepAtom model [58] and others [54, 59–64]. Note that these new models usually employ a large training set with extra data from general sets from PDBbind. Details of the training sets, testing sets, and performance (PCC) of these models are listed in Table 5.

**Docking power.** We test the docking power, which is to identify the native poses from the ones generated by docking softwares [65], of our model on benchmark CASF-2013. There are totally 195 testing ligands in CASF-2013, each ligand has 100 poses generated from three docking programs, GOLD v5.1, Surflex-Dock in SYBYL v8.1 and MOE v2011. A pose is considered to be a native one if its RMSD value with respect to the true binding pose is less than 2 Å. Detailed RMSD information of all the ligands can be found in CASF-2013. If the pose with the highest predicted binding energy is a native one, it is regarded as a successful prediction. Once

**Table 3. The PCCs and RMSEs ($pK_d/pK_i$) for our DC-GBT models in three test cases, i.e., PDBbind-v2007, PDBbind-v2013 and PDBbind-v2016.** Three DC-GBT models are considered with features from different types of bipartite graphs. The DC-GBT(Dist) model uses features from distance-based bipartite graphs; The DC-GBT(Chrg) model uses features from electrostatic-based bipartite graphs; The DC-GBT(Dist+Chrg) model uses features from both distance-based bipartite graphs and electrostatic-based bipartite graphs.

| Dataset | Dist | Chrg | Dist+Chrg |
|---|---|---|---|
| PDBbind-v2007 | 0.816(1.416) | 0.811(1.437) | 0.824(1.402) |
| PDBbind-v2013 | 0.789(1.457) | 0.790(1.456) | 0.799(1.432) |
| PDBbind-v2016 | 0.836(1.270) | 0.834(1.284) | 0.843(1.255) |
| Average | 0.813(1.377) | 0.812(1.385) | 0.822(1.357) |

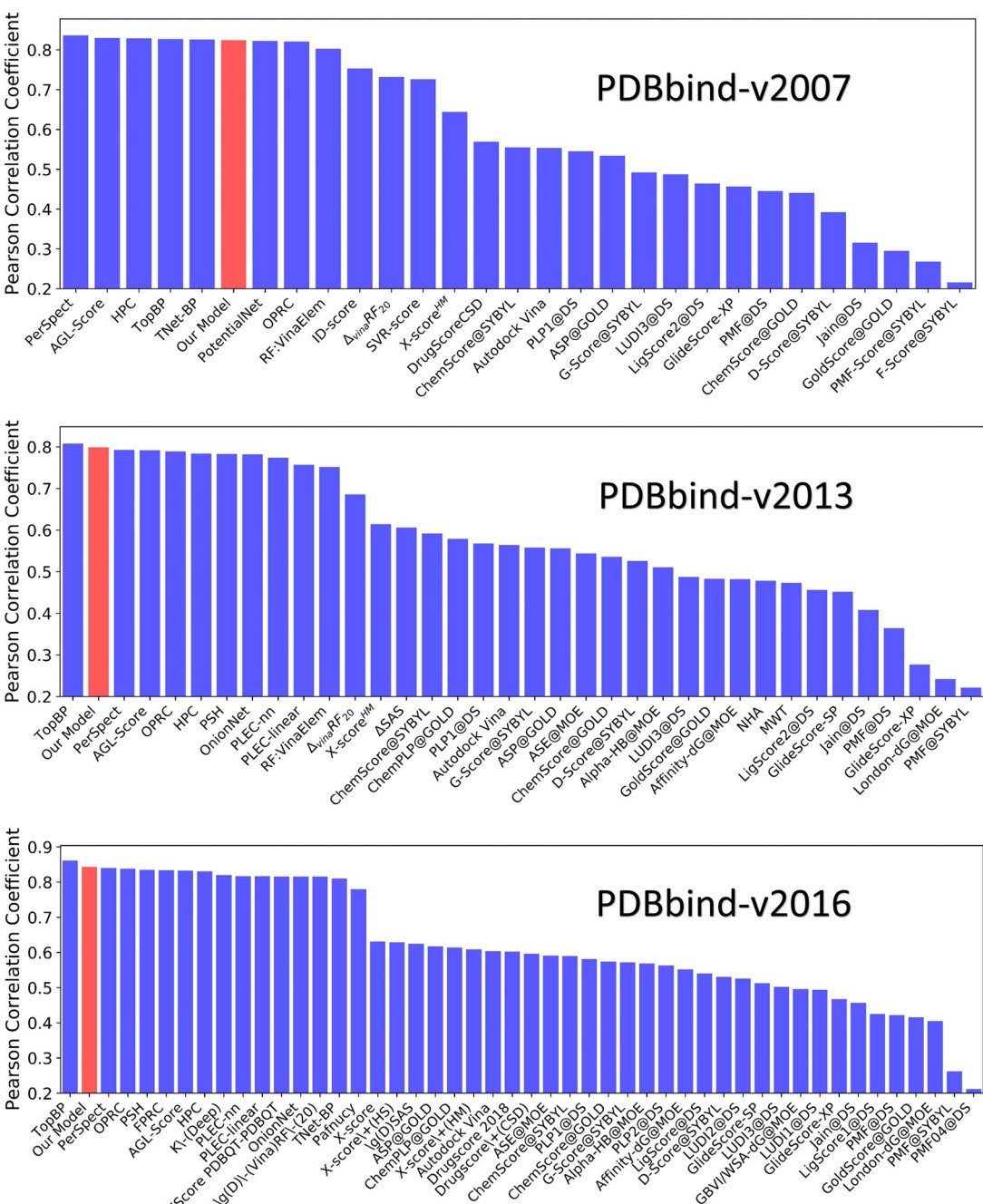

**Fig 3. Preformance comparison between our models and other models.** The comparison of PCCs between our model and other molecular descriptor based models, for the prediction of protein-ligand binding affinity. The PCCs are calculated based on the core set (test set) of PDBbind-v2007, PDBbind-v2013 and PDBbind-v2016.

this process is performed for the whole 195 testing ligands, an overall success rate can be computed for the given scoring function.

For each ligand, an individual training and testing process is needed. We repeat our DC-GBT(Dist) model for each of the 195 ligands, following the procedure in the work [18].

**Table 4. The comparison of our DC-GBT model with advanced-mathematical based machine learning models [14, 18, 20, 30, 32, 33].** Note that values marked with * uses PDBbind-v2016 core set ($N$ = 290).

| Model | PDBbind-v2007 | PDBbind-v2013 | PDBbind-v2016 | Average |
|---|---|---|---|---|
| AGL-Score | 0.830 | 0.792 | 0.833 | 0.818 |
| HPC-GBT | 0.829 | 0.784 | 0.831 | 0.815 |
| TNet-BP | 0.826 | N/A | 0.810* | N/A |
| TopBP | 0.827 | **0.808** | **0.861*** | **0.832** |
| PerSpect | **0.836** | 0.793 | 0.840 | 0.823 |
| OPRC | 0.821 | 0.789 | 0.838 | 0.816 |
| DC-GBT | 0.824 | 0.799 | 0.843 | 0.822 |

Note that GOLD v5.6.3 [66] is used to generate 1000 training poses for each ligand. These poses and their scores are available at https://weilab.math.msu.edu/AGL-Score.

In our implementation, ten independent regressions are performed for each ligand. The ligand is regarded as a successful one if at least three regressions successfully identify the native poses. In this case, our success rate can reach 88%. If we use at least six successful regressions as standard, our success rate drops slightly to 86%.

**Screening power.** The screening power of a scoring function is its ability to identify the true binders for a given target protein from decoy structures. We test our model on benchmark CASF-2013. There are totally 65 different proteins in CASF-2013. For each protein, there are at least three true binders while the rest of the 195 ligands are regarded as decoys. There are two kinds of screening power measurements. One is to find out the enrichment factors (EF) among the $x$% top-ranked molecules.

$$EF_{x\%} = \frac{number\ of\ true\ binders\ among\ x\%\ top\ ranked\ molecules}{(total\ number\ of\ true\ binders\ for\ the\ given\ protein) \times x\%}$$

where top-ranked molecules means the predicted candidates with high binding energies. And the average EF value among all 65 proteins is used to assess the screening power of a scoring function. The other is the success rate to identify the best true binders. For each target protein, if the the best binders are found in $x$% top-ranked molecules, this protein is taken as a successful one. Then the overall success rate is given by the total number of successful proteins over 65.

For each protein, an individual training and testing process is considered. Our DC-GBT (Dist) model is used for each of 65 proteins, following the works [18]. More specifically, for a

**Table 5. The performance in terms of PCCs and RMSEs ($pK_d/pK_i$) for recently-proposed models using different training sets [54–64].** Note that values marked with * uses PDBbind-v2016 core set ($N$ = 290), and the values marked with + uses PDBbind-v2013 core set($N$ = 180) and PDBbind-v2016 core set($N$ = 276).

| Model | Training set | Testing set1 core(PDB-v2013) | Testing set2 core(PDB-v2016) |
|---|---|---|---|
| graphDelta | PDB-v2018(8766) | | 0.87(1.05) |
| ECIF | PDB-v2019(9299) | | 0.866(1.169) |
| OnionNet-2 | PDB-v2019(>9000) | 0.821(1.357) | 0.864(1.164) |
| DeepAtom | PDB-v2018(9383) | | 0.831(1.232)* |
| Ligand-based | PDB-v2018(11663) | 0.780(N/A)+ | 0.821(N/A)+ |
| SE-OnionNet | PDB-v2018(11663) | 0.812(1.692) | 0.83(N/A) |
| DeepDTAF | PDB-v2016(11906) | | N/A(1.443)* |
| Deep Fusion | PDB-v2016(9226) | | 0.803(1.327*) |

given target protein, AutoDock Vina is used to dock all the ligands in PDBbind-v2015 refined set, excluding the core set and the true binders of this protein. This procedure gives rise to a few thousand of training poses and associated energy labels for each target protein. The binding scores (kcal/mol) generated by AutoDock Vina is converted to binding energy ($pK_d$) by multiplying a constant -0.7335. Those binders in the refined set that do not bind to the target protein are regarded as decoys, and their binding energies should be smaller than the true binders. Therefore, if the energy of a decoy generated by AutoDock Vina is higher than the lower bound of the true binders' energies, the decoy is relabeled by the lower bound of the true binders' energies. Note that this procedure may cause many different decoys sharing same labels. The binding energy of a protein-ligand complex is usually a positive value, so we exclude the entries with negative binding energies from the training sets for all the 65 protein cases. The poses and associated scores can be found at https://weilab.math.msu.edu/AGL-Score.

In our implementatoin, for each protein, ten independent regressions are performed. As in PDBBind dataset, two decimal places are kept for the predicted binding energies of 195 testing ligands. For each regression, an EF value can be obtained and whether this protein is a successful one can be judged. The average EF value of ten regressions is taken as the final EF value for this protein in our model. Note that there are many different entries in the training set sharing the same labels, which results in that our model may predict same binding energies for different ligands. In this case, we take the first and second ranked ligands as the 1% top-ranked candidates. Hence the number of 1% top-ranked candidates maybe larger than 2. In this case, we set the EF to be 66.6. For the overall success rate, if at least two cases among the ten regressions assert that the given protein is a successful one, this protein is regarded as a successful one. In this case, our success rate can reach to 68%. If at least six successful regressions are used as standard, our success rate drops to about 58%.

Note that in our DC-GBT model, we consider target-specific scoring models and train them separatively for scoring, docking and screening power tests. Previously, general scoring function models were developed. That is for all the tasks, the same general scoring function is used. Recently, data-driven learning models make use of the different types of training datasets to improve the performance of scoring functions. In particular, the incorporation of decoy data in the training set can significantly improve the docking and screening power [67–72]. In our DC-GBT models, different training sets are used, which results in different state-of-the-art scoring models in scoring, docking and screening power tests.

## Discussion

Machine learning models have made tremendous progresses in text, video, audio and image data analysis. In particular, convolutional neural network (CNN) models have achieved revolutionary advancements in the analysis of image data. However, molecular data from material, chemical and biological systems are fundamentally different from text and image data, as their properties are usually directly determined by their topological structures. Persistent models, including persistent homology/cohomology, persistent functions, and persistent spectral, provide a series of highly effective molecular descriptors that not only preserve intrinsic structural information, but also maintain molecular multiscale properties. Here we propose Dowker complex based machine learning models for drug design. Dowker complex is used for molecular interaction representation. Riemann Zeta functions are defined on persistent spectral models and further used as molecular descriptors. Our Dowker complex based machine learning models have achieved state-of-the-art results for protein-ligand binding affinity. They can be also be used in AI-based drug design and other molecular data analysis.

## Materials and methods

Our model contains two essential components, i.e., DC-based molecular representation and DC-based PerSpect models. For a molecular interaction-based bipartite graph, its associated DCs can be decomposed into two disjointed DCs which have the same homology groups. The DC-based Hodge-Laplacian matrices and Riemann Zeta function can be constructed and further used in the generation of molecular features for machine learning models.

### DC-based persistent homology

Dowker complex is based on the "relations" of two sets and is originally developed to explore the homology of relations [38]. Mathematically, a relation is equivalent to a bipartite graph. So for each bipartite graph, a Dowker complex can be naturally constructed. More specifically, let $G$ be a bipartite graph, the DC $K_D(G)$ will have two disjoint components $K_{D,1}(G)$ and $K_{D,2}(G)$. Assume $G = (V_1, V_2, E)$ is a connected bipartite graph where $V_1$ and $V_2$ are two vertex sets and $E$ is the edge set that form only between $V_1$ and $V_2$. $K_D(G) = K_{D,1}(G) \cup K_{D,2}(G)$ where $K_{D,1}(G)$ and $K_{D,2}(G)$ are the two disjoint components of $K_D(G)$, $K_{D,1}(G)$ and $K_{D,2}(G)$ are defined as follows:

- $K_{D,1}(G)$: for a set of vertices $\{x_{i_0}, x_{i_1}, \ldots, x_{i_p}\}$ in $V_1$, a $p$-simplex is formed among these vertices in $K_{D,1}(G)$, if there exists a vertex $y \in V_2$ such that $\{(x_{i_m}, y)|0 \leqslant m \leqslant p\} \subset E$.

- $K_{D,2}(G)$: for a set of vertices $\{y_{i_0}, y_{i_1}, \ldots, y_{i_p}\}$ in $V_2$, a $p$-simplex is formed among these vertices in $K_{D,2}(G)$, if there exists a vertex $x \in V_1$ such that $\{(x, y_{i_m})|0 \leqslant m \leqslant q\} \subset E$.

An example can be found in Fig 4. It can be seen that the Dowker complex just has two disjoint components, one is in black points and the other is in green points, and a simplex is formed if their vertices have a common neighbor vertex in the bipartite graph. Like the triangle in black points, its three vertices has a common green point as neighbor vertex in the bipartite graph. We have $H_p(K_{D,1}(G)) \cong H_p(K_{D,2}(G))(0 \leqslant p)$. Actually $K_{D,1}(G)$ and $K_{D,2}(G)$ are homotopic equivalent.

Finally, if we construct the bipartite graph filtration process as in Eq (2), and induce the DC-based filtration as in Eq (3). Two separated DC sequences are generated as Eqs (4) and (5). The corresponding persistent barcodes of these two DC sequences are exactly same.

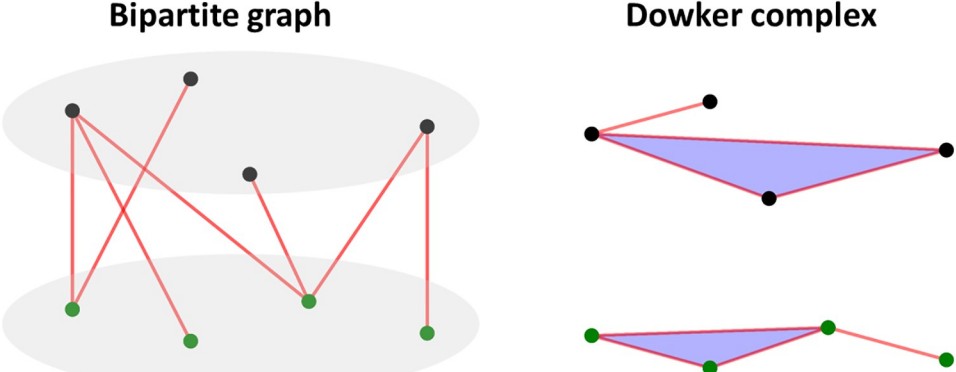

**Fig 4. A bipartie graph and its associated Dowker complex.** It can be seen that there are two disjoint components in DC, one is from the black points and the other is from the green points. Note that a triangle (2-simplex) is formed among the black point set in DC, as the corresponding three black vertices have a common neighbor blue vertex in the bipartite graph.

## DC-based PerSpect models

Persistent spectral theory studies the spectral evolutional information of combinatorial Laplacian matrixes associated with a filtration process. An oriented DC is needed for the construction of combinatorial Laplacian matrixes, but different orientations share the same eigen spectral information. In this way, we can define an orientation based on the sequence of atoms (as in PDB file) for simplicity.

For an oriented DC $K_D = \{\delta_k^i; k = 0, 1, \ldots; i = 1, 2, \ldots\}$, its $k$-th boundary matrix $B_k$ can be defined as follows,

$$B_k(i,j) = \begin{cases} 1, & \text{if } \delta_i^{k-1} \subset \delta_j^k \text{ and } \delta_i^{k-1} \sim \delta_j^k \\ -1, & \text{if } \delta_i^{k-1} \subset \delta_j^k \text{ and } \delta_i^{k-1} \not\sim \delta_j^k \\ 0, & \text{if } \delta_i^{k-1} \not\subset \delta_j^k. \end{cases}$$

Here $\delta_i^{k-1} \subset \delta_j^k$ means that $\delta_i^{k-1}$ is a face of $\delta_j^k$ and $\delta_i^{k-1} \not\subset \delta_j^k$ means the opposite. The notation $\delta_i^{k-1} \sim \delta_j^k$ means the two simplexes have the same orientation, i.e., oriented similarly, and $\delta_i^{k-1} \not\sim \delta_j^k$ means the opposite.

The $k$-th laplacian matrix is defined as follows,

$$\mathbf{L}_k = \mathbf{B}_k^T \mathbf{B}_k + \mathbf{B}_{k+1} \mathbf{B}_{k+1}^T.$$

More specifically, $L_0$ can be expressed explicitly as,

$$L_0(i,j) = \begin{cases} d(\delta_i^0), & \text{if } i = j \\ -1, & \text{if } i \neq j, \delta_i^0 \frown \delta_j^0, \\ 0, & \text{if } i \neq j, \delta_i^0 \not\frown \delta_j^0 \end{cases} \tag{8}$$

Further, $L_k(k > 0)$ can be expressed as,

$$L_k(i,j) = \begin{cases} d(\delta_i^k) + k + 1, & \text{if } i = j \\ 1, & \text{if } i \neq j, \delta_i^k \not\frown \delta_j^k, \delta_i^k \smile \delta_j^k \text{ and } \delta_i^k \sim \delta_j^k \\ -1, & \text{if } i \neq j, \delta_i^k \not\frown \delta_j^k, \delta_i^k \smile \delta_j^k \text{ and } \delta_i^k \not\sim \delta_j^k \\ 0, & \text{if } i \neq j, \delta_i^k \frown \delta_j^k \text{ or } \delta_i^k \not\smile \delta_j^k. \end{cases}$$

Here $d(\delta_i^k)$ is (upper) degree of $k$-simplex $\delta_i^k$. It is the number of $(k + 1)$-simplexes, of which $\delta_i^k$ is a face. Notation $\delta_i^k \frown \delta_j^k$ means the two simplexes are upper adjacent, i.e., they are faces of a common $(k + 1)$-simplex, and $\delta_i^k \not\frown \delta_j^k$ means the opposite. Notation $\delta_i^k \smile \delta_j^k$ means the two simplexes are lower adjacent, i.e., they share a common $(k - 1)$-simplex as their face, and $\delta_i^k \not\smile \delta_j^k$ means the opposite. Notation $\delta_i^k \sim \delta_j^k$ means the two simplexes have the same orientation, i.e., oriented similarly, and $\delta_i^k \not\sim \delta_j^k$ means the opposite.

In our model, we consider the Riemann Zeta functions on the spectral as our persistent attributes. More specifically, for a set of eigenvalues $\{\lambda_1, \lambda_2, \ldots, \lambda_n\}$, the spectral moment of the simplicial complex can be defined as the Zeta function $\zeta(s) = \sum_{i=1}^n \frac{1}{\lambda_i^s}$. Then we use the persistent spectral moment as the persistent attributes for machine learning.

## Author Contributions

**Conceptualization:** Kelin Xia.

**Data curation:** Xiang Liu, Kelin Xia.

**Formal analysis:** Xiang Liu, Kelin Xia.

**Funding acquisition:** Kelin Xia.

**Investigation:** Kelin Xia.

**Methodology:** Xiang Liu, Kelin Xia.

**Project administration:** Kelin Xia.

**Resources:** Huitao Feng, Jie Wu, Kelin Xia.

**Supervision:** Kelin Xia.

**Validation:** Xiang Liu, Kelin Xia.

**Visualization:** Xiang Liu.

**Writing – original draft:** Kelin Xia.

**Writing – review & editing:** Xiang Liu, Huitao Feng, Jie Wu, Kelin Xia.

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
