## [Decision Letter · Decision Letter 0]

12 Oct 2021

Dear Dr. Xia,

Thank you very much for submitting your manuscript "Dowker complex based machine learning (DCML) models for protein-ligand binding affinity prediction" for consideration at PLOS Computational Biology.

As with all papers reviewed by the journal, your manuscript was reviewed by members of the editorial board and by several independent reviewers. In light of the reviews (below this email), we would like to invite the resubmission of a significantly-revised version that takes into account the reviewers' comments.

We cannot make any decision about publication until we have seen the revised manuscript and your response to the reviewers' comments. Your revised manuscript is also likely to be sent to reviewers for further evaluation.

Sincerely,

Joanna Slusky, Ph.D.

Associate Editor

PLOS Computational Biology

Arne Elofsson

Deputy Editor

PLOS Computational Biology

Reviewer's Responses to Questions

**Comments to the Authors:**

Reviewer #1: Here the authors proposed a Dowker complex based molecular interaction representations, which used a bipartite graph to model the interactions between a protein and a ligand. Then a DC-based persistent spectral model was constructed and the persistent Riemann Zeta functions were calculated as molecular descriptors. Finally, a DC-based gradient boosting tree model was trained to predict protein-ligand binding affinity.

This is a novel method to represent protein-ligand interaction as bipartite graph and calculate the descriptors from the knowledge of topology and graph theory. When it was applied to protein-ligand affinity prediction, however, I have some concerns about the representations and models:

1. In order to calculate the descriptors, the binding core region was defined using a cutoff distance of 10A. I wonder how you defined the cutoff. Actually, I have seen some different definitions about the binding core region, ranging from 5A to 12A. Does the cutoff distance influence the results a lot?

2. According to the manuscript, the size of feature vectors depend on the filtration values and the number of Riemann Zeta functions. Did they have physical or mathematical significance? Or were they selected by hyper-parameter optimization?

3. In page 8/17, line 226: “Note that the accuracy of our DC-based models can be further improved if convolutional neural network models, such as the one used in TopBP models…” Have you already tried the convolutional neural network models or you just imagined that?

4. The Table 1 listed the detailed information of the three PDBBind databases. I noticed that the Training set includes all the remained data when removing Test set from Refined set. Is there a validation set when you train your model? And how the hyper-parameters listed in Table 2 were selected?

5. There are many different type of protein-ligand affinity prediction models, which can also be called scoring functions. The scoring power is not the only problem we concerned, there are test sets for testing the docking power and screening power in CASF-2016 (or other version). We are very interested in the docking power and screening power of the model. We suggested that you provide the related results.

6. In page 8/17, line 231: “We do not compare with these models because the training and testing sets of these models are different from the standard ones in PDBbind datasets” Considering that all the PDBBind datasets are public, it is not difficult to make a comparison. I think more evidence should be given to prove the advantage of DC-based molecular interaction representations.

Reviewer #2: This work proposes novel molecular descriptors for protein-ligand binding affinity predictions. These descriptors are constructed from Dowker complex and spectral graph information. The authors have validated the robustness and the efficiency of the proposed features against series of PDBbind benchmarks. Overall this manuscript is well-written and easy to follow. Besides these positive sides, there are some downsides I would like to bring up here

1) Proposed models use charges, distances, DC-based features, etc. General readers will appreciate it if authors carefully investigate the performances of the separated features. There might be some redundant features.

2) I do not know how atom charges were obtained. Please provide such a discussion in the revised version.

3)Lines 226-228, authors claim that CNN can further improve their current model. Are there any hard proofs? If yes please provide them otherwise I suggest removing these sentences.

3) Please include TopBP in figure 3 since it is discussed in Table 4

4) There are missing data files/features files from the authors’ provided Github link. Please update them.

**Have the authors made all data and (if applicable) computational code underlying the findings in their manuscript fully available?**

Reviewer #1: None

Reviewer #2: **No: **Missing data files/feature files

PLOS authors have the option to publish the peer review history of their article (what does this mean?). If published, this will include your full peer review and any attached files.

Reviewer #1: No

Reviewer #2: No
---

## [Editor Report · Decision Letter 1]

6 Jan 2022

Dear Dr. Xia,

Thank you very much for submitting your manuscript "Dowker complex based machine learning (DCML) models for protein-ligand binding affinity prediction" for consideration at PLOS Computational Biology.

As with all papers reviewed by the journal, your manuscript was reviewed by members of the editorial board and by several independent reviewers. In light of the reviews (below this email), we would like to invite the resubmission of a significantly-revised version that takes into account the reviewers' comments.

After reviewing your new manuscript I noted that a significant number of the clarifications and requests by reviewers resulted in responses to the reviewers that did not yield changes to the manuscripts. Please consider the reviewers as representatives of your broader audience. Almost anything on which the reviewer needed clarification future readers will need clarification as well. Therefore, if a concept or detail needs to be explained to the reviewers, it also needs to be explained to the audience of PLoS Computational Biology in the manuscript. I cannot send this revision back to reviewers until you have added your responses to the manuscript as well.

In addition, it would be helpful to add quotes to the response to reviewers with the precise language you use in the manuscript to address the reviewers concerns. This allows the reviewers and editor to find your changes more easily and see in context how you addressed previous concerns.

We cannot make any decision about publication until we have seen the revised manuscript and your response to the reviewers' comments. Your revised manuscript is also likely to be sent to reviewers for further evaluation.

Sincerely,

Joanna Slusky, Ph.D.

Associate Editor

PLOS Computational Biology

Arne Elofsson

Deputy Editor

PLOS Computational Biology

After reviewing your new manuscript I noted that a significant number of the clarifications and requests by reviewers resulted in responses to the reviewers that did not yield changes to the manuscripts. Please consider the reviewers as representatives of your broader audience. Almost anything on which the reviewer needed clarification future readers will need clarification as well. Therefore, if a concept or detail needs to be explained to the reviewers, it also needs to be explained to the audience of PLoS Computational Biology in the manuscript. I cannot send this revision back to reviewers until you have added your responses to the manuscript as well.

In addition, it would be helpful to add quotes to the response to reviewers with the precise language you use in the manuscript to address the reviewers concerns. This allows the reviewers and editor to find your changes more easily and see in context how you addressed previous concerns.
---

## [Decision Letter · Decision Letter 2]

31 Jan 2022

Dear Dr. Xia,

Thank you very much for submitting your manuscript "Dowker complex based machine learning (DCML) models for protein-ligand binding affinity prediction" for consideration at PLOS Computational Biology.

As with all papers reviewed by the journal, your manuscript was reviewed by members of the editorial board and by several independent reviewers. In light of the reviews (below this email), we would like to invite the resubmission of a significantly-revised version that takes into account the reviewers' comments.

We cannot make any decision about publication until we have seen the revised manuscript and your response to the reviewers' comments. Your revised manuscript is also likely to be sent to reviewers for further evaluation.

Sincerely,

Joanna Slusky, Ph.D.

Associate Editor

PLOS Computational Biology

Arne Elofsson

Deputy Editor

PLOS Computational Biology

Reviewer's Responses to Questions

**Comments to the Authors:**

Reviewer #1: The authors have responded adequately to most of my comments, but there are some problems in the response to the question about docking power and screening power. The scoring, docking and screening powers should be evaluated using the same scoring function model, but the authors retrained their model for each ligand in the docking power test and each protein in the screening power test. The performance of these target-specific scoring models cannot be compared to the performance of those general scoring functions listed in Figure 4, except the AGL-Score. In other words, the model which is used to evaluate the docking power and screening power should be the same model that is used to evaluate scoring power.

Reviewer #2: The authors have addressed all of my concerns.

**Have the authors made all data and (if applicable) computational code underlying the findings in their manuscript fully available?**

Reviewer #1: Yes

Reviewer #2: Yes

PLOS authors have the option to publish the peer review history of their article (what does this mean?). If published, this will include your full peer review and any attached files.

Reviewer #1: No

Reviewer #2: No
---

## [Editor Report · Decision Letter 3]

21 Feb 2022

Dear Dr. Xia,

We are pleased to inform you that your manuscript 'Dowker complex based machine learning (DCML) models for protein-ligand binding affinity prediction' has been provisionally accepted for publication in PLOS Computational Biology.

Best regards,

Joanna Slusky, Ph.D.

Associate Editor

PLOS Computational Biology

Arne Elofsson

Deputy Editor

PLOS Computational Biology

---

## [Editor Report · Acceptance letter]

22 Mar 2022

PCOMPBIOL-D-21-01678R3 

Dowker complex based machine learning (DCML) models for protein-ligand binding affinity prediction

Dear Dr Xia,

I am pleased to inform you that your manuscript has been formally accepted for publication in PLOS Computational Biology. Your manuscript is now with our production department and you will be notified of the publication date in due course.

With kind regards,

Livia Horvath
